# Targeting Protein Misfolding and Aggregation as a Therapeutic Perspective in Neurodegenerative Disorders

**DOI:** 10.3390/ijms252212448

**Published:** 2024-11-20

**Authors:** Marta Sidoryk-Węgrzynowicz, Kamil Adamiak, Lidia Strużyńska

**Affiliations:** Laboratory of Pathoneurochemistry, Department of Neurochemistry, Mossakowski Medical Research Institute, 02-106 Warsaw, Poland; kadamiak@imdik.pan.pl (K.A.); lidkas@imdik.pan.pl (L.S.)

**Keywords:** protein aggregation, neurodegeneration, antibody, fulvic, dynasore, anle138b, epigallocatechin gallate

## Abstract

The abnormal deposition and intercellular propagation of disease-specific protein play a central role in the pathogenesis of many neurodegenerative disorders. Recent studies share the common observation that the formation of protein oligomers and subsequent pathological filaments is an essential step for the disease. Synucleinopathies such as Parkinson’s disease (PD), dementia with Lewy bodies (DLB) or multiple system atrophy (MSA) are neurodegenerative diseases characterized by the aggregation of the α-synucleinprotein in neurons and/or in oligodendrocytes (glial cytoplasmic inclusions), neuronal loss, and astrogliosis. A similar mechanism of protein Tau-dependent neurodegeneration is a major feature of tauopathies, represented by Alzheimer’s disease (AD), corticobasal degeneration (CBD), progressive supranuclear palsy (PSP), and Pick’s disease (PD). The specific inhibition of the protein misfolding and their interneuronal spreading represents a promising therapeutic strategy against both disease pathology and progression. The most recent research focuses on finding potential applications targeting the pathological forms of proteins responsible for neurodegeneration. This review highlights the mechanisms relevant to protein-dependent neurodegeneration based on the most common disorders and describes current therapeutic approaches targeting protein misfolding and aggregation.

## 1. Introduction

Although neurodegenerative diseases differ in their clinical, pathological, and biochemical characteristics, they share a common feature related to the misfolding and aggregation of pathological, disease-specific proteins, followed by the death of neurons [1]. Disease-specific misfolded protein/s found in the brain include, for example, amyloid-beta (Aβ) and hyperphosphorylated Tau protein in Alzheimer’s disease (AD), α-synuclein in Parkinson’s disease (PD), multiple system atrophy (MSA), the prion protein (PrP) in prion diseases, and SOD1 in Lou Gehrig’s disease. Pathological protein deposits in the cytoplasm of motor neurons are a molecular feature of amyotrophic lateral sclerosis (ALS). In this disorder, the most common protein aggregations are represented by the excessive deposition of TAR DNA-binding protein 43 (TDP-43) [2]. The two major proteins that aggregate in the frontotemporal dementia (FTD) are Tau and TDP-43, whereas a minority of patients aggregate FET proteins and the FUS protein [3]. Among neurodegenerative disorders are also polyglutamine (polyQ) diseases, such as Huntington’s disease (HD), the onset of which has been linked to abnormally expanded CAG trinucleotide repeat coding for the polyQ sequence in mutant Huntingtin proteins. The mutant proteins are unrelated except for the polyQ tract. It is well-established that aggregated polyQ is a major component of the deposits that are found in diseased brains [4,5] (Table 1).

Most of these diseases are sporadic, but some are inherited in a dominant manner. Mutations frequently occur in the genes encoding Aβ, Tau, and α-synuclein, giving rise to disease-associated phenotypes. Protein assembly begins in specific regions of the brain during Alzheimer’s and Parkinson’s diseases and spreads to other areas as the disease progresses [20]. There is substantial in vivo evidence for the intracerebral cell-to-cell transfer of protein aggregates, also known as prion-like spreading, causing neurodegeneration. For example, several studies have shown that the intracerebral injection of pathological α-synuclein fibrils into transgenic or wild-type mice resulted in neurodegeneration [21]. Most neurodegenerative diseases involve the spread of degeneration between individuals and among cells or tissues via template-directed misfolding, in which misfolded protein conformers mediate disease by causing normal proteins to misfold [22].

Recent studies, including many high-profile drug trials, have directly targeted these disease-specific misfolded proteins [23]. Furthermore, the experimental approaches are supported by numerous preclinical studies. This review focuses on describing the most common specific proteins involved in neurodegenerative processes and then presents novel approaches targeting pathological proteins as highly promising therapeutic strategies for the treatment of neurodegenerative disorders.

## 2. Protein Misfolding and Aggregation as Common Features of Neurodegeneration: Examples of the Most Common Disorders

### 2.1. Tau-Dependent Neurodegeneration

Tau is a highly expressed soluble phospho-protein that is present in neurons in both the central (CNS) and peripheral nervous systems (PNS), with its highest representation found in axonal processes [24,25]. Normally, Tau is enriched in the axons, where it interacts with microtubules (MTs) and may regulate MT polymerization and dynamics. Tau plays a fundamental role in proper neuronal functioning as is involved in the assembly and stabilization of microtubules (MTs), neurodevelopment, axonal maintenance, and transport [26,27]. However, the pathological aggregation of Tau into hyperphosphorylated filaments mediates the transformation of the protein into paired helical filament (PHF) structures, which results in MTs depolymerization and Tau dissociation, followed by the formation of more complex neurofibrillary tangles (NFTs). NFTs composed of the aggregates of hyperphosphorylated Tau protein are widely known as a primary biomarker of diseases collectively called tauopathies [28,29]. Although Tau is present in dendritic spines under physiological conditions [30,31], the infiltration of the pathological form of Tau can disrupt synaptic functions and lead to synapse loss [32,33]. These pathological changes mediate the disruption of axonal transport and synaptic transmission and cause neuronal death. Thus, abnormal Tau aggregation is a defining feature of tauopathies, including Alzheimer’s disease (AD) and corticobasal degeneration (CBD), progressive supranuclear palsy (PSP), argyrophilic grain disease (AGD), globular glial tauopathy (GGT), and Pick’s disease (PD) [4,34]. Human genetics established the link between protein dysfunction and neurodegeneration. A dominantly inherited form of frontotemporal dementia and parkinsonism was found to be associated with chromosome 17q21–22, the region where *MAPT* is located [35]. Furthermore, mutations in *MAPT* led to the production of transgenic rodent lines that exhibit neurodegeneration relevant to Tau pathology [36,37]. It is well-established that mutations in *MAPT* are relevant to a type of frontotemporal dementia associated with parkinsonism [38]. The importance of Tau oligomers, filaments, and pathological protein formation in neuronal dysfunction is supported by the correlation of the distribution and severity of Tau pathology with the clinical phenotypes of tauopathies.

### 2.2. Amyloid β-Related Pathology

The genetic background of Alzheimer’s disease differs from early-onset familial Alzheimer’s disease, other cases of early-onset Alzheimer’s disease, and late-onset Alzheimer’s disease. Rare cases of early-onset familial Alzheimer’s diseases are caused by mutations in genes coding for amyloid precursor proteins, namely, presenilin 1 and presenilin 2, while late-onset Alzheimer’s disease is more complex and is associated with different genetic risk loci, with the apolipoprotein E ε4 allele being a major genetic disease risk factor [39,40]. AD is a characterized by the abnormal aggregation and deposition of amyloid-β (Aβ) peptides into extracellular plaques and of hyperphosphorylated Tau into intracellular neurofibrillary tangles, followed by synaptic and neuronal loss resulting in progressive cognitive and functional decline. Senile plaques formed by the extracellular amyloid-β peptide represent major hallmarks of AD pathology [41]. Aβ is a small, soluble peptide produced from the amyloid precursor protein (APP) after sequential cleavage mediated by the enzymes β-secretase and γ-secretase to form the N-terminus and C-terminus, respectively. The cleavage releases the soluble amino-terminal APP fragments, whereas imbalances between the Aβ production and clearance lead to the accumulation of Aβ, which can spontaneously aggregate into soluble oligomers or insoluble fibrillar structures that ultimately accumulate into senile plaques. Aβ_40_ are the most abundant forms of secreted Aβ, whereas insoluble amyloid deposits found in AD brains consist of the neurotoxic isoform Aβ_42_ [42]. Recent studies have shown that soluble Aβ oligomers, rather than insoluble fibrillar aggregates or plaques, are responsible for synaptic loss and cognitive impairment [43]. Moreover, these deficits preceded the appearance of insoluble Aβ aggregates in the transgenic PD mice model, suggesting that AD pathophysiology occurs prior to plaque deposition and that oligomers should be considered as a target for therapeutic intervention [44].

### 2.3. α-Synucleinopathies

The molecular characterization of PD patients and families led to the identification of several disease-associated genes. Some genes are associated with a Mendelian inheritance pattern, while others elevate the risk of PD progress with ageing. There are several genes related to the monogenic PD (e.g., *SNCA*, *LRRK2*, *PRKN*, *PINK1*, *VPS35*, and *VPS13C*) and major genetic risk factors (e.g., *GBA1*, and *LRRK2*) [45]. αSyn consists of 140 amino acids highly expressed in the brain, coding by gene SNCA, which resides in chromosome 4 [46]. Several studies demonstrated that α-synuclein gene mutations located in the N-terminal region (A30P, E46K, A53T, H50Q, and G51D) are crucial for this disease. Additional information regarding mutations, as well as relevant studies that discuss the role of mutations, is described in Table 2.

Structurally, α-synuclein is represented by three regions: the N-terminal part containing seven imperfect repeats of 11 amino acids with a highly conserved KTKEGV motif and with propensity to form alpha-helical structures [60]. The primary site of αSyn localization is the synapse, and this protein is mainly implicated in neurotransmitter homeostasis. This homeostasis appears by the regulation of synaptic vesicle fusion and trafficking between the restored and releasing pools, as well as by interplay with membrane transporters of the neurotransmitter. αSyn natively exists as an unfolded monomer and does not have a secondary structure. Under pathological conditions, αSyn tends to be misfolded and aggregated into multiple soluble oligomeric species and also into insoluble amorphous or fibrillar amyloid-like assembles. It is well-established that the appearance of the aggregates of misfolded α-synuclein is a common feature of PD. αSyn forms filaments that represent neuropathological lesions found in patients with Parkinson’s disease (PD), dementia with Lewy bodies (DLB), multiple system atrophy (MSA), and other progressive neurodegenerative disorders collectively known as α-synucleinopathies [61,62]. The major feature of these diseases is the formation of intracellular inclusions composed of filamentous aggregated protein in neuronal (e.g., Lewy bodies, Lewy neurites) and glial populations (oligodendrocytes, cytoplasmic inclusions). Several studies have shown that the intracerebral injection of α-synuclein fibrils (e.g., synthetic fibrils or fibrils taken from the brain samples of patients with Parkinson’s disease, dementia with Lewy bodies, or MSA) into transgenic or wild-type mice resulted in neurodegeneration [63,64]. It is well-accepted that αSyn aggregation within the synapses, causing alterations in synaptic structure and function, sustains the primary event that initiates the pathogenesis of α-synucleinopathies. Therefore, inhibition of α-synuclein aggregation is a therapeutic approach targeting a key pathophysiological processes in synucleinopathies [65].

## 3. Pathological Proteins and Neurodegeneration

### 3.1. Common Pattern of Protein Aggregation

The folding of newly synthesized polypeptide chains into their native conformations is crucial for achieving a specific functional state of the protein. However, as a result of the partial or complete unfolding of the polypeptide chain, monomeric proteins undergo misfolding and aggregation, leading to the formation of various aggregated structures such as amyloid aggregates with β-sheets running parallel to the long axis of the fibrils (a structure known as cross-β). The soluble aggregates formed during the protein aggregation generally known as oligomers are heterogeneous and can rapidly interconvert into large, toxic fibrillar aggregates (protofibrils) in the diseased brain (Figure 1). αSyn is considered to natively exist as an unfolded monomer with little secondary structure; studies report the formation of αSyn tetramers and higher order multimers like octamers under physiological conditions. Under pathological conditions, αSyn undergo misfolding and aggregation into multiple soluble oligomeric species and eventually become insoluble amorphous or amyloid like protofibrils [66,67,68]. The precise mechanisms leading to pathological Tau aggregation are not fully established. As compared to amyloid aggregation, the development of Tau pathology exhibits more complexity and variability. When Tau becomes pathological, the protein undergoes different post-translational modifications such as glycosylation, cleavage, nitration, ubiquitination, glycation, or hyperphosphorylation [69]. Ongoing post-translational modifications in the microtubule-binding domains have been proposed to initiate Tau aggregation by first weakening its interaction with negatively charged microtubules. The reduced affinity of Tau toward the microtubules upon these modifications elevate the template of soluble, modified Tau molecules that might have a higher tendency to self-assemble into aggregates [70,71].

### 3.2. Common Mechanisms of Protein Spreading

A growing body of evidence suggests that disease-specific protein aggregates are capable of self-sustaining amplification and spreading along neuronal connections, thereby leading to disease progression from specific brain regions in the early phase of the disease to widespread areas in its advanced stages [Figure 2]. Once a misfolded form of the protein enters a neighboring cell, it can act as a template for the misfolding of the natively folded form of the protein, leading to the loss of function and efficient propagation of the disease. This pathway is commonly referred to as ‘prion-like’ spreading. Tau, Aβ, or α-Syn aggregates, appearing in the target cell, can cause the formation of additional protein aggregates, leading to the amplification of pathological proteins and subsequent cell damage and death [72]. Protein aggregates released from donor cells are taken up by neighboring cells, thus spreading from cell to cell.

The glial cell pool in the CNS consists of macroglia (astrocytes and oligodendrocytes) and microglia, the immune cells of the brain. Astrocytes are the most abundant cells, with extraordinary properties involved in a wide variety of brain functions, including maintenance of the blood–brain barrier, fluid and ionic balance, and regulation of the neuronal synapses [73]. A highly intricate network of cellular communication comprises the complex interactions of the neurons and glial cells that are involved in the propagation of pathological insults. Studying the underlying molecular mechanisms involved in seeding, amplification, and spreading may thus lead to the identification of promising therapeutic targets [23]. Another important issue relevant to therapeutic strategies includes the inhibition of protein aggregation and misfolding. Here, we focus on agents with different chemical properties and mechanisms of action that rescue specific protein pathology by affecting their misfolding.

## 4. Novel Approaches to In Vitro Analysis of Protein Aggregation

The insolubility of protein aggregates in mild detergents and their partial resistance to protease digestion are thought to play a role in the development of neurodegeneration, so investigating which drugs can prevent pathological aggregation could lead to the development of more effective treatments. Originally established in prion-related diseases, the protein misfolding cyclic amplification (PMCA) technique combines cycles of incubation to grow fibrils and sonication to break the fibrils into smaller growing fractions. It has been found to be useful for the rapid amplification of accumulated proteins that exhibit biophysical and biochemical characteristics similar to the aggregates present in patient brain inclusions. PMCA-produced alpha-synuclein aggregates have a high content of β-sheet structures, demonstrate partial resistance to proteinase K digestion, and are filamentous. Finally, recent studies demonstrated that the use of PMCA is suitable for screening drugs affecting α-synuclein by monitoring the spread and internalization of protein aggregates in vitro [74].

Moreover, several other types of assays have been developed to visualize proteins and thus monitor their interactions or cell-to-cell transferring in biological models. For example, protein fragment complementation assays (PFCAs) use a fluorescent protein or an enzyme that has been truncated and fused to two proteins of interest. When these two proteins interact with each other, both complementary fragments lead to the reconstitution of the reporter activity, using specific enzymatic or fluorescent properties [75,76]. One great example of the PFCA method is bimolecular fluorescence complementation (BiFC), which is based on the reconstitution of a fluorescent protein when it is linked to different proteins. This convenient approach allows results to be obtained without additional modifications to the experimental models. BiFC is suitable for monitoring protein release and transfer during the co-culturing of cells expressing proteins of interest fused to each of the non-fluorescent fragments of the fluorescently marked target protein [77].

The disappearance of monomers and the formation of early oligomers in a concentration-dependent manner, accompanied by a conformational change that precedes β-structure formation, can be visualized by fluorescence resonance energy transfer (FRET) [78,79]. For example, FRET methodology was adapted for studying α-Syn pathology, where the labeled mutant α-synuclein was a fluorescent probe with trace amounts in the presence of excess unlabeled α-synuclein. Another useful approach to study protein aggregation kinetics and localization patterns in cell translocation is pulse shape analysis (PulSA), a flow cytometry-based method. Examples for its use include tracking the formation of the inclusion bodies of polyglutamine-expanded proteins and other aggregating proteins [80]. Another example of aggregated protein visualization is stimulated Raman scattering (SRS) microscopy. This technology provides high chemical specificity for endogenous biomolecules and can circumvent common constraints of fluorescence microscopy and facilitates the analysis of native polyQ aggregates [79]. Another important issue is that the proteins associated with neurodegenerative disease possess a high propensity to condense via the mechanism of liquid–liquid phase separation (LLPS), and once the phase is separated, the proteins are transitioned into liquid condensates, favoring misfolding and aggregation [81]. For example, a recent study revealed that the liquid–liquid phase separation of α-Syn appears in the nucleation event of α-Syn aggregation, which offers this alternate, non-canonical aggregation pathway [81]. Furthermore, Tau droplet formation by liquid–liquid phase separation may be the initial step in Tau aggregation, leading to results that disrupt the role of Tau in dendritic functions [81].Therefore, LLPS inhibition provides a promising strategy against pathological protein misfolding and aggregation.

## 5. Pharmacological Interventions in Protein Aggregation and Internalization

### 5.1. The Effect of anle138b on Pathological Proteins

Recent evidence indicates tremendous progress in developing promising pharmaceutical agents and provides a better understanding of the pathology of neurodegenerative diseases.

A great example of this is the study of small organic agents targeting pathological oligomer aggregation [82]. For example, diphenyl–pyrazole (DPP) compounds, represented by diphenyl–pyrazole anle138b (3-(1,3-benzodioxol-5-yl)-5-(3-bromophenyl)-1H-pyrazole), have been identified as new pharmaceutical compounds for the treatment of diseases dependent on specific protein aggregation. This agent was selected after screening a primary library containing drug-like compounds and performing subsequent biochemical optimization using a variety of methods, including a novel method of scanning for intensely fluorescent targets (SIFT) [83]. Anle138b does not bind to the monomer, which is a great advantage from a therapeutic point of view, because this drug is unable to interfere with the physiological functions of the non-aggregated protein. Anle138b has been demonstrated to be an effective drug that modulates toxic oligomers by allowing high-affinity binding to a structural epitope responsible for misfolding along the amyloidogenic pathway. The specific properties of this compound allows binding that destabilizes toxic oligomers, impedes the formation of oligomer pores in membranes, and prevents the prion-like propagation of the synuclein species [84,85]. Anle138b exhibits structure-dependent binding to aggregates that effectively disrupts the formation of pathological oligomers in vitro and in vivo, including α-synuclein, prion protein, Aβ, and Tau [86]. Notably, anle138b is characterized by high oral bioavailability and efficient penetration of the blood–brain barrier (BBB), and these properties are the reason for its efficacy and higher concentration in the brain than in the plasma without the signs of toxicity at therapeutic doses that were present in an animal model [86].

In vitro studies revealed the ability of anle138b to block the formation of the pathological aggregates of prion protein and αSyn [83]. A study using HEK293 revealed that anle138b reduces αSyn aggregation in living cells [87].

Growing evidence indicates that anle138b potently inhibits oligomer accumulation, neuronal degeneration, and disease progression in vivo in various mouse models [88,89]. An example of this is an in vivo study using a well-established murine model of α-synucleinopathy {(Thy1)-h[A30P]α-syn} with a genetic background of C57/Bl6 mice. In this case, the animals were clinically evaluated on several criteria such as survival time, motor performance, and body weight and exhibited a significant reduction in the disease progression by anle138b [90].

PLP-hα-Syn mice expressing human α-synuclein in oligodendrocytes were rescued from motor dysfunction after the oral administration of anle138. The histological and molecular analyses showed a significant decline in αSyn oligomer aggregation and glial cytoplasmic inclusions, which was accompanied by reduced microglial activation in the substantia nigra region [88].

Another experimental approach revealed that late-stage anle138b treatment effectively reduced the Tau content in the frontal cortex and the hippocampi of mice expressing all six human Tau isoforms and rescued metabolic function. Molecular brain imaging with longitudinal 18F-fluorodeoxyglucose positron emission tomography (FDG-PET) showed restored cerebral metabolic function in the frontal cortexes of transgenic mice administered with anle138b [91].

A study using mice overexpressing P301S mutant human Tau revealed that anle138b binds to aggregated Tau and inhibits Tau aggregation both in vitro and in vivo. Another consequence of using anle13b is decreased synapse and neuronal loss and reduced gliosis in the CA3 region of the mouse hippocampus. Moreover, anle138b effectively reverses disease symptoms and ameliorates the survival times and cognitive functions of transgenic animals [83].

A previous study showed that treatment with the oligomer modulator anle138b, started during the symptomatic phase of the disease at 50 weeks of age, has a significant impact on surviving a terminal disease in a PD mouse model [90]. Martinez Hernandez et al. [92] showed that anle138b also interferes with Aβ toxicity. They first observed that anle138b was able to improve survival in a Drosophila model of amyloid-induced neurotoxicity. This result encouraged the authors to analyze its effect in a mouse model of AD (APPPS1ΔE9), in which Aβ deposits began to appear at 6 months of age and later (8–10 months of age), when some neuronal loss was observed around amyloid plaques. Four months of treatment with anle138b improved long-term potentiation (LTP), the molecular response to learning and memory processes, in the hippocampi of the mice. Consistently, anle138b treatment was associated with an improved spatial memory in the Morris water maze test, as compared to the impaired spatial memory seen in animals with the same genetic backgrounds but treated with a placebo. Anle138b improved memory in a mouse model of AD and did not affect the cognitive functions of the control mice. These results were demonstrated in mice both before the onset of the pathology and after the onset of amyloid deposition at the pre-plaque (2-month-old) and post-plaque (6-month-old) stages, respectively. These results strongly suggest that anle138b is a potent inhibitor of amyloid-induced toxicity with potential therapeutic properties. RNA sequencing of the affected pathways in the hippocampi of wild-type and APPPS1ΔE9 mice at the pre-plaque stage revealed several alterations in genes (203 genes) related to cell growth, energy metabolism, mitochondrial function, cytoskeleton, and synaptic plasticity. Interestingly, most of these alterations were reversed by the administration of anle138b, suggesting that the overall effect of these gene changes correspond to the cellular functions.

### 5.2. EGCG and Protein Pathology

Epigallocatechin gallate (EGCG) is a catechin present in green tea that exhibits anti-amyloidogenic properties and modulates the misfolding of disease proteins and prions [93]. EGCG directly binds to unfolded polypeptide chains and inhibits β-sheet formation, an early event in the amyloid formation cascade, suggesting that this agent redirects aggregation-prone polypeptides to protein assemblies located off the pathway [94]. These findings have encouraged researchers to study whether EGCG might also be able to disassemble pre-formed β-sheet-rich structures as well as earlier intermediates of fibrillogenesis. A study on the human neuroblastoma cell-line model revealed that EGCG reduced α-Syn aggregation [95]. EGCG has been shown to exert anti-amyloid activity by influencing cellular signal transduction pathways and reactive oxygen species [93]. An in vitro study demonstrated that EGCG has the ability to remodel mature alpha-synuclein species into smaller, amorphous protein aggregates that are relatively nontoxic to mammalian cells. EGCG exposure was found to enhance the clearance of AD-relevant phosphorylated Tau species in primary neurons [96].

EGCG administration also prevents Tau aggregation by binding the Tau phosphorylation region and modifying the 3D-structure of Tau [97]. Fluorescence assay with Thioflavin-S (ThS) demonstrated the impeding effect of EGCG on an aggregation of Tau. The 8-anilino-1-naphthalenesulfonic acid (ANS) fluorescence showed a time-dependent decrease in protein intensity, suggesting EGCG-mediated effects on the hydrophobicity of Tau aggregates. The same experimental approach revealed the involvement of EGCG in dissolving processes of preformed Tau filaments and oligomers. Notably, the incubation of neuroblastoma cells with EGCG led to the formation of rather nontoxic Tau, and the reversed Tau toxicity was accompanied by enhanced neuronal growth and survival [98].

### 5.3. Fulvic Acid-Mediated Effects

Fulvic acid (FA) is a mixture of various polyphenolic acids present in humans with high antioxidant properties and neuroprotective activity [99]. It is well-established that FA interacts with prion protein, negatively affecting the content of β-sheet structures and the formation of protein aggregates [87]. Fulvic acid was found to promote the disaggregation of Tau protein in cells cultured under non-proliferative conditions [100]. A previous study using HEK293 demonstrated that FA not only inhibits heparin-induced Tau aggregation in vitro but also promotes the disassembling of preformed Tau fibrils in living cells [87]. Interestingly, another group reported that FA can also increase neurite outgrowth and, when used in combination with a B-vitamin complex, may improve cognitive functions in AD patients [101]. These studies suggest that FA, as well as the FA formulation with B vitamins, is emerging as a novel nutraceutical source with potential use in the treatment of neurodegenerative disorders.

### 5.4. Dynasore and Protein Pathology

Endocytosis is dependent on dynamin for the invagination of the plasma membrane to form clathrin-coated pits, and dynamin polymerizes to form a helix around the neck of the budding vesicles of the plasma membrane, leading to membrane fission and the generation of free clathrin-coated vesicles. Dynasore is a GTPase inhibitor that rapidly and reversibly inhibits dynamin activity, thereby preventing endocytosis [102]. Dynasore is a non-competitive, reversible inhibitor of dynamin1 and dynamin2 that interferes with the catalytic step of the protein’s GTPase activity, leaving GTP bound to dynamin2 and dynamin1.

It is well-established that α-Syn interacts with various membrane proteins [103] and regulates the dopamine D2 receptor’s function, resulting in internalization via the caveolae-mediated endocytic pathway. A study using primary dopaminergic neurons demonstrated that the disruption of caveolae by dynasore, as well as caveolin-1 knockdown, abolished the uptake of α-Syn monomers by introducing primary cultures of dopaminergic neurons [104].

Reyes et al. [105] established a model of N2a neuroblastoma cells transfected with plasmid vectors expressing C-terminally tagged αSyn, allowing for the quantification of cell-to-cell transfer of αSyn, detected by flow cytometry and high-content analysis. This approach revealed that dynasore deregulates the transfer and spreading of αSyn between cells. Another in vitro study focusing on the cell-to-cell propagation of αSyn demonstrated that oligodendrocytes take up recombinant α-Syn monomers, oligomers, or fibrils, and that this process is inhibited by dynasore [105].

Recent studies have demonstrated multiple roles for dynasore in neurodegeneration that are not just limited to α-synuclein-related pathology. For example, dynasore was also found to reduce Tau pathology by affecting protein internalization in the HEK293 cell line [87].

### 5.5. Antibody-Mediated Therapy

#### 5.5.1. Tauopathies

Active and passive vaccines are two widely accepted immunotherapy strategies for the treatment of neurodegeneration disorders. Active immunization involves the administration of the pathogenic agent via injection, while passive immunization relies on the action of a specific antibody to target a given antigen. Immunotherapy targeting the pathological protein has also been extensively studied in Tau-related neurodegeneration. The first Tau vaccine tested in human is AADvac1, which consists of Axon peptide 108 (N-terminally cysteinylated Tau 294-305/4R, amino acid sequence CKDNIKHVPGGGS) coupled to keyhole limpet haemocyanin (KLH). Immunization with this agent successfully eliminated the major signs of neurofibrillary pathology and resulted in a significant clinical improvement of the transgenic rat population [106]. The efficacy and safety of the AADvac1 vaccine, currently in phase 2 clinical trials, were assessed after administration to a population of people with mild Alzheimer’s disease and to a parallel placebo group [107].

The liposome-based vaccine ACI-35, consisting of 16 copies of a synthetic Tau fragment phosphorylated at S396 and S404, is a good example of the active Tau immunotherapy already used in clinical trials. ACI-35 has been reported to reduce the clinical parameters tested, including extension of lifespan or the slowing down of the clasping motor phenotype in mice [108].

Another experimental study demonstrated that the administration of three effective antibodies into the lateral ventricle of P301S mice significantly reduced the hyperphosphorylation, aggregation, and insolubility of Tau in the brain lysates and prevented the development of Tau seeding. Moreover, this approach effectively reduced microglial activation and improved the cognitive functions of the experimental animals, suggesting that specifically designed immunotherapy alleviates disease symptoms from this initial point to more advanced stages, which is relevant to the intercellular propagation of protein aggregates [109].

More recently, an unbiased proteomic analysis of postsynaptic densities (PSDs) in a Tau-P301S transgenic mouse model demonstrated Tau-relevant pathology in synapses prior to overt neurodegeneration. This analysis allows us to identify the alteration of multiple proteins and pathways in the postsynaptic fraction of Tau-P301S. The pathway analysis, together with molecular approaches, revealed that the specific depletion of postsynaptic GTPase-regulating proteins and the dysregulation of the actin cytoskeleton constitute the primary mechanisms of dendritic spine loss and pathology. It is well-established that the complement components of the innate immune system are expressed and secreted by glial cells, ensuring the clearance of damaged cells and synapse removal [110]. A study revealed the elevation of the extracellular complement component C1q in the PSDs of Tau-P301S mice and in AD patients correlated with Tau pathology. Dejanovic et al. [111] showed that Tau pathology induces the engulfment of the excitatory synapses by microglia in a C1q-dependent manner and that the main consequence of microglia action is a decline of synapse density leading to actin cytoskeletal disruption, followed by the loss of dendritic spines. This finding encouraged the authors to use a C1q-neutralizing antibody as a specific target to prevent microglia-mediated synapse removal. This approach revealed that the use of the C1q antibody prevents synapse engulfment by microglia and restores synaptic function in both neuronal cultures and the Tau-P301S mouse model.

Albert et al. [112] demonstrated that the anti-Tau D antibody, which recognizes an epitope in the central region of Tau, effectively neutralizes the pathological Tau species in the brains of P301L Tg mice previously injected with human AD brain extracts. Targeting the mid-domain of Tau with antibody D prevents the appearance of neurofibrillary degeneration in the pyramidal neurons of the CA1 region. Furthermore, the injection of Tau fibrils, which are not recognized by the anti-Tau antibody D in transgenic mice, resulted in blocking the progression of Tau pathology to the distal brain regions.

In turn, passive immunization with the Tau antibody against the N-terminal domain of Tau not only reduced Tau pathology but also ameliorated Aβ pathology in triple-transgenic (3×Tg)-AD mice in moderate to severe stages of the disease [113]. The same research group found that passive immunization with the monoclonal antibody 43D against the proximal N-terminal domain of Tau could reduce both Tau and Aβ pathology in 3×Tg-AD mice and that the N-terminal proximal domain of Tau was a more effective target for passive immunization than the N-terminal distal domains of Tau [113]. Another study also revealed immunotherapeutic effects against disease progression in the diseased brain. Sevingy et al. [114] addressed this issue by using a monoclonal antibody (Aducanumab) directed against neurotoxic Aβ aggregates in humans suffering from AD. AD patients who received Aducanumab showed a significant reduction in the number of amyloid plaques observed in PET scans and slower cognitive decline [114], suggesting that therapeutic interventions targeting Aβ could be beneficial. Aducanumab is a human antibody that selectively binds to aggregated forms of Aβ, including insoluble fibrils and soluble oligomers. A study in Tg2576 mice (APPSwe) showed that the acute topical application of the murine analog of Aducanumab to the animal brains resulted in the clearance of existing amyloid plaques. In addition, this study revealed that immunotherapy is more effective in the prevention or treatment of amyloidosis in earlier stages but is less effective in advanced stages where there are substantial parenchymal plaque deposits [115].

In general, active and passive immunotherapies targeting different phospho-Tau peptides have great potentials in several transgenic mouse models of AD [116]. Active immunization with a phosphorylated form of Tau reduces its pathology and reverses or slows cognitive decline in rodents [117,118,119], while passive immunotherapy using antibodies against Tau has also been shown to slow disease progression [108].

#### 5.5.2. Synucleinopathies

As mentioned above, recent studies demonstrated that specific antibodies directed against Tau and Aβ protein-mediated pathology may be useful for preventing disease development and/or progression. Similarly, several studies have revealed the usefulness of monoclonal antibodies in αSyn-mediated pathology and disease-related deficits [120].

Given that the C-terminal (CT) truncation of α-Syn is crucial for the pathogenesis of PD, several studies have focused on passive immunization against this protein region. One example of this approach is a study on the mThy1-αSyn transgenic mouse model resembling the striatonigral and motor deficits of PD. Monoclonal antibodies 1H7, 5C1, or 5D12, directed against the CT of α-Syn, caused the synaptic and axonal pathology in immunized animals to decline by decreasing the levels of CT-αSyn and higher molecular-weight aggregates, as well as by decreasing the content of CT-truncated α-syn in axons.

In addition to these effects, they prevented the loss of tyrosine hydroxylase fibers in the striatum and improved motor and cognitive deficits in animals exposed to the antibodies, with 1H7 and 5C1 having the strongest effects. Furthermore, in vitro studies showed that the preincubation of recombinant αSyn with 1H7 and 5C1 prevented the CT cleavage of α-Syn, which seems to be important for protein misfolding and cell-to-cell propagation [121].

Another study focusing on the immunization strategy to prevent αSyn-related pathology revealed a significant role for α-Syn extracellular clearance, mediated by microglia. Co-cultures consisting of primary microglia and a conditioned medium containing αSyn released from the SH-SY5Y neuronal cell line revealed a specific anti-synuclein, antibody-dependent clearance of aggregates by microglia. Studies using the BV-2 microglia cell line model demonstrated that the antibody-dependent clearance of α-Syn aggregates is based on FCγ receptor-mediated protein internalization and its delivery to the lysosomal system. Furthermore, the unilateral injection of αSyn antibody reduced neuronal and glial accumulation of α-Syn in the cortex of the hippocampi of the Tg mouse model of α-synucleinopathies. Finally, passive immunization by the intraperitoneal injection of antibodies to animals affected with neurodegenerative pathology showed improved functional motor coordination [122].

Another example of an immunotherapeutic strategy is the use of the high-affinity anti-α-Syn antibody MEDI1341 in PD pathology. These antibodies were found to enter the brain, interact with αSyn, and dysregulate αSyn spreading in vivo. Notably, MEDI1341 possesses affinity to monomeric, oligomeric, and aggregated inclusions composed of α-Syn. MEDI1341 also inhibits the intercellular translocation of pre-formed αSyn fibrils in vitro. A study with a lentivirus-based mouse model of αSyn entering the brain showed that the addition of MEDI1341 significantly rescued axons from αSyn accumulation and spreading [123].

## 6. Conclusions

A growing body of evidence suggests a common mechanism dependent on the improper folding and function of specific proteins associated with the neurodegenerative processes, disease pathology, and its progression. Previous studies have demonstrated that a specific drug or immunization targeting the pathological factor reduces protein accumulation and disease-related abnormalities. The progress described here, elucidating the molecular pathogenesis of neurodegenerative disorders, offers new approaches to the development of effective therapies that, in some cases, are in the phase of human clinical trials.

## Figures and Tables

**Figure 1 ijms-25-12448-f001:**
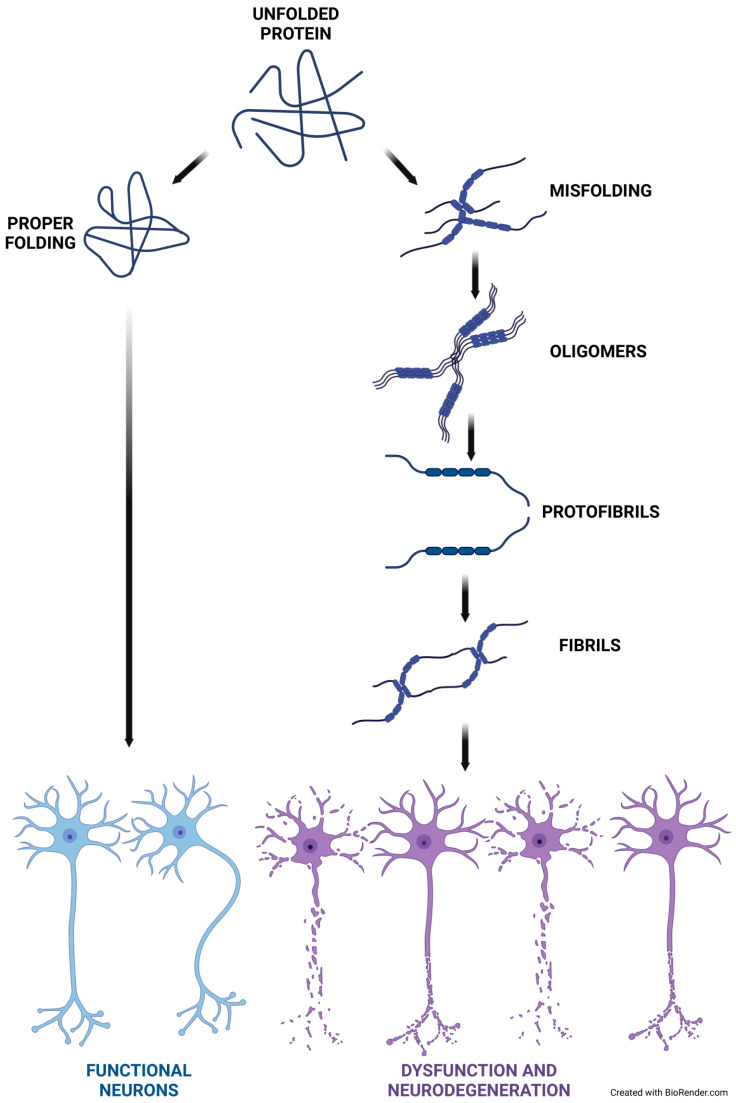
Protein misfolding and neurodegeneration. Many etiologically unrelated neurodegenerative disorders manifest a common pathology like the accumulation of insoluble aggregates. Under pathological conditions, diseases with specific naive proteins undergo misfolding and aggregation in the form of filamentous amyloid deposits, leading to neuronal dysfunction. Created in BioRender. Pan, I. (2024) https://BioRender.com/u01m506 (accessed on 13 November 2024).

**Figure 2 ijms-25-12448-f002:**
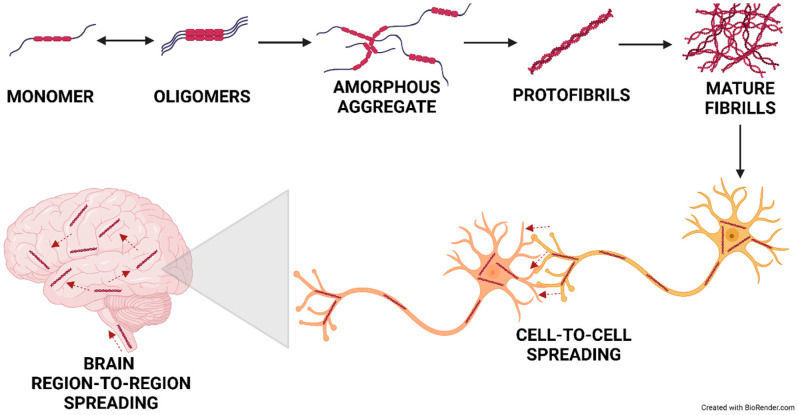
Mechanism of protein translocation in neurodegeneration. The spreading of protein misfolding operates at different levels during neurodegeneration, including molecule-to-molecule, cell-to-cell, and region-to-neighbor in a brain region and between interconnected brain regions. Created in BioRender. Pan, I. (2024) https://BioRender.com/a72r749 (accessed on 13 November 2024).

**Table 1 ijms-25-12448-t001:** Misfolding and aggregation of specific proteins and their relevance to neurodegenerative disorders.

Disease	Protein	Length	Dominant Aggregation Mechanism	References
Alzheimer’s disease (AD)	Amyloid-β (Aβ) Tau (3R + 4R)	40–42 352–441	Secondary nucleation	[6,7]
Parkinson’s disease (PD)	α-synuclein	140	Condition-dependent, including lipid-induced aggregation	[8,9]
Dementia with Lewy Bodies (DLB)	α-synuclein	140	Not clear, secondary pathways	[10,11]
PD dementia (PDD)	α-synuclein	140	Not clear, possibly secondary pathway	[12]
Multiple system atrophy (MSA)	α-synuclein	140	Not yet known, possibly secondary pathway	[13]
Pick’s disease (PiD)	Tau (3R)	352–410	Not yet known, possibly secondary pathway	[14]
Corticobasal degeneration (CBD)	Tau (4R)	383–441	Not yet known, possibly secondary pathway	[15]
Progressive supra- nuclear palsy (PSP)	Tau (4R)	383–441	Not yet known, possibly secondary pathway	[16]
Argyrophilic grain disease (AGD)	Tau (4R)	383–441	Not yet known, possibly secondary pathway	[16]
Globular glial tauopathy (GGT)	Tau (4R)	383–441	Not yet known, possibly secondary pathway	[16]
Spongiform encephalopathies	Prion protein (PrP)	208	Fragmentation	[17,18]
Huntington’s disease (HD)	Huntingtin	variable	Not yet known, possibly oligomers	[19]

**Table 2 ijms-25-12448-t002:** Examples of mutations associated with protein-dependent neurodegeneration.

Protein	Mutation	References
Aβ	E693G	[47,48]
Tau	G272V	[49]
	P301L	[38]
	V279M	[50]
	V337M	[50]
	R406W	[51]
α-Syn	A53T	[52]
	A53E	[53]
	A30P	[54]
	E46K	[55]
TDP-43	A315T	[56]
	Q331K	[57]
	M337V	[57]
	D169G	[58,59]

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
