# Peer review of "Targeting Protein Misfolding and Aggregation as a Therapeutic Perspective in Neurodegenerative Disorders"

_ijms, 2024, doi:10.3390/ijms252212448_

Round 1

Reviewer 1 Report

Comments and Suggestions for Authors

This is a review of one of medicine's main topics: neurodegeneration. The review updates existing knowledge and provides an almost complete perspective, from proteinopathies to therapies, including nice figures and an informative table.
As weak points, 3 aspects could be treated in more detail:
- the genetic background of neurodegenerative diseases
- copathology and related challenges in diagnostics and therapeutics
- gene therapies in research, including ASOs, CRISPR/Cas....

Author Response

Comments 1:  the genetic background of neurodegenerative diseases
Response 1: Thank you for this suggestion. In the current version we are providing additional informations about gnentic backgroud of described disorders (sections: 2.1; 2.2; 2.3) with the relevant references.

Comments 2:- copathology and related challenges in diagnostics and therapeutics 

Response 2: We agree. This is import and broad and issue in  the field of human neurodegeneration. A great example for it is AD pathology where  abnormal aggregation of amyloid-β (Aβ) peptides and extracellular plaques coexist with  hyperphosphorylated Tau/intracellular neurofibrillary tangles. We adding these  information in the section 2.2. However, copatology might be tricky for the cure. Thefore it is needed to target primary disease related protein for sufficient treatment and this review focuses on the most effective/known approaches. Ineed, this keeps our work more consistent.

Comments 3: - gene therapies in research, including ASOs, CRISPR/Cas

Response 3: Thank you very much for this suggestion. In case of neurodegeneration the use of  gene therapy  is quite complicated, because disorers are mostly idiopathic. There are some ASOs approaches in case of Huntington Disease, but without spectacular outputs. The second  thing is that our work, although touches on many topics, is focused on the pathology associated with protein aggregation.  

Reviewer 2 Report

Comments and Suggestions for Authors

The review paper by Sidoryk-Węgrzynowicz et al. reviewed the recent progresses of recognizing the role of protein misfolding in neurodegenerative disorders, and highlighted efforts of pharmacological interventions to mitigate protein misfolding and aggregation as means to slow down disease progression. Therapeutic efforts to tackle neurodegeneration is a hot research area, therefore this review is timely, and also suits the scope of IJMS. However, I do have some suggestions which may help to strengthen the scientific soundness of this paper.

1.        As the authors introduced Tau, a-syn and PrP in great length, other relevant proteins are completely missing. These include SOD1, TDP43, FUS, Htt-PolyQ etc. The aggregation of these proteins is involved in many neurodegenerative diseases including ALS and Parkinson’s disease. Therefore, these proteins are relevant to the scope of this review, and should be included in section 2.

2.        I recommend the authors to include a figure that has the linear representation of all these disease-relevant proteins, where the readers could easily access to some of the important information, including protein length, domains, mutations, etc.

3.        A few methods were published to report the neurodegeneration-relevant protein misfolding in live cells, they should be discussed in the section 4. In particular, fluorogenic tools that harbors microenvironment detection dyes were reported to detect and distinguish misfolded protein oligomers and aggregates (a review could be found at PMID:35040316). Other methods including using Stimulated Raman Scattering microscopy (PMID: 32341997), flow cytometry (PMID: 22426490), FCS/FRET (PMID:20371330). These works should be briefly discussed in section 4.

4.        Another big advancement in this field has been connecting protein liquid-liquid phase separation and misfolding. Once phase separated, proteins are transitioned into liquid condensates which have much higher concentration, therefore favoring misfolding and aggregation. Drugs inhibiting LLPS has also been discussed to slow down disease progression of neurodegeneration. I felt like a brief discussion should be included at the end.

Author Response

Comments1: As the authors introduced Tau, a-syn and PrP in great length, other relevant proteins are completely missing. These include SOD1, TDP43, FUS, Htt-PolyQ etc. The aggregation of these proteins is involved in many neurodegenerative diseases including ALS and Parkinson’s disease. Therefore, these proteins are relevant to the scope of this review, and should be included in section 2.

Response 1: We agree with the Reviewer that this important issue was missed. Current version is enriched with  examples of other diseases proteins/mediators  (SOD1, TD43, FUS, Htt-PolyQ; Section 1)

Comments 2: I recommend the authors to include a figure that has the linear representation of all these disease-relevant proteins, where the readers could easily access to some of the important information, including protein length, domains, mutations, etc.

Response 2: Thank you for this suggestion. We included  Huntington  Disease to the Table 1 with suitable  reference. We included  additional Table 2 with informations regarding mutations, as well as relevant studies that discuss the role of mutations/diseases. We belive that this will allow readers to easily find important informations.

Comments 3: A few methods were published to report the neurodegeneration-relevant protein misfolding in live cells, they should be discussed in the section 4. In particular, fluorogenic tools that harbors microenvironment detection dyes were reported to detect and distinguish misfolded protein oligomers and aggregates (a review could be found at PMID:35040316). Other methods including using Stimulated Raman Scattering microscopy (PMID: 32341997), flow cytometry (PMID: 22426490), FCS/FRET (PMID:20371330). These works should be briefly discussed in section 4.

Response 3: We agree that additional methods for neurodegeneration-relevant protein detection will significantly improve our manuscript. We enriched current version with all mentioned by Reviewer tools to detect pathogenic species (FRET, flow cytometry,Stimulated Raman Scattering; Section 4) with relevant references disscusting these approaches.

Comments 4: Another big advancement in this field has been connecting protein liquid-liquid phase separation and misfolding. Once phase separated, proteins are transitioned into liquid condensates which have much higher concentration, therefore favoring misfolding and aggregation. Drugs inhibiting LLPS has also been discussed to slow down disease progression of neurodegeneration. I felt like a brief discussion should be included at the end.

Response 4: Thank you, this is very important suggestion and  worth to be mention. In the Section 4. we briefly discuss  this issue and present current publications focusing the protein liquid-liquid phase separation and missfolding